# ExplainCPE: A Free-text Explanation Benchmark of Chinese Pharmacist Examination

**Dongfang Li**[*], **Jindi Yu**[*], **Baotian Hu**[†], **Zhenran Xu** and **Min Zhang**

Harbin Institute of Technology (Shenzhen), Shenzhen, China

crazyofapple@gmail.com

{hubaotian, zhangmin2021}@hit.edu.cn

{22S051013, xuzhenran}@stu.hit.edu.cn

## Abstract

In the field of Large Language Models (LLMs), researchers are increasingly exploring their effectiveness across a wide range of tasks. However, a critical area that requires further investigation is the interpretability of these models, particularly the ability to generate rational explanations for their decisions. Most existing explanation datasets are limited to the English language and the general domain, which leads to a scarcity of linguistic diversity and a lack of resources in specialized domains, such as medical. To mitigate this, we propose ExplainCPE, a challenging medical dataset consisting of over 7K problems from Chinese Pharmacist Examination, specifically tailored to assess the model-generated explanations. From the overall results, only GPT-4 passes the pharmacist examination with a 75.7% accuracy, while other models like ChatGPT fail. Further detailed analysis of LLM-generated explanations reveals the limitations of LLMs in understanding medical text and executing computational reasoning. With the increasing importance of AI safety and trustworthiness, ExplainCPE takes a step towards improving and evaluating the interpretability of LLMs in the medical domain. The dataset is available at https://github.com/HITsz-TMG/ExplainCPE.

## 1 Introduction

Advancements in the field of Large Language Models (LLMs) (Radford et al., 2019; Brown et al., 2020; Ouyang et al., 2022), exemplified by models such as GPT-4 (OpenAI, 2023), have opened up new possibilities and challenges across a myriad of natural language processing (NLP) tasks (Wei et al., 2022a). These models have shown remarkable success in understanding and generating human-like text, promoting research that spans a wide array of applications (Bubeck et al., 2023).

A critical aspect that remains unexplored is the interpretability of these models, specifically the ability to provide accurate and faithful rationale for their decisions (Wei et al., 2022b; Turpin et al., 2023). The degree to which these models can explain their reasoning is of crucial significance, especially in high-stakes domains such as healthcare, where the clarity of generated responses directly impacts decision-making and outcomes (Rudin, 2019).

Current benchmarks for LLMs mostly focus on the exam performance, such as MMLU (Hendrycks et al., 2021) and AGIEval (Zhong et al., 2023). These datasets do not allow for a detailed assessment of LLMs' justifications of their decisions, because of the unavailability of high-quality and professional explanations. Moreover, accurately measuring the explainability of these LLMs is a difficult task due to the lack of comprehensive and standardized datasets that come from unbiased and trustworthy sources (Li et al., 2023). Existing benchmarks predominantly are from online forums and consumer feedback and only consist of English-language general knowledge questions (Wiegreffe and Marasovic, 2021), which results in insufficient thematic and linguistic diversity. Overall, the lack of appropriate evaluation datasets has prevented a full understanding of LLMs' strengths and weaknesses in the field of interpretability.

To address this gap, we introduce ExplainCPE, a challenging medical benchmark dataset in Chinese, encompassing over 7K instances. This dataset, specifically tailored to evaluate the capacity of model explainability, diversifies the linguistic scope of interpretability research and allows for a rigorous assessment of model performance in a specialized, high-stakes domain. An example from our dataset is presented in Table 1. The in-depth analysis of LLMs performance on ExplainCPE brings to light several critical observations. First,

---

[*]Both authors contributed equally to this work.
[†]Corresponding author.

| |
|---|
| **Question** |
| The patient, female, 65 years old, received the second cycle of chemotherapy for breast cancer. Which antiemetic drugs are prohibited? |
| **Options** |
| **A.**Metoclopramide.✓   **B.**Ondansetron.   **C.**Dexamethasone.   **D.**Diphenhydramine.   **E.**Aprepitant. |
| **Gold Explanation** |
| This question examines antiemetic drugs. Optional antiemetic drugs are mainly dopamine receptor blockers (such as metoclopramide), 5-HT3 receptor blockers (such as ondansetron, tropisetron, and palonosetron, etc.) , corticosteroids (such as dexamethasone), anticholinergics and antihistamines (such as diphenhydramine), and NK-1 receptor blockers (such as aprepitant), etc. Special emphasis should be placed on the prohibition of metoclopramide as an antiemetic in patients with breast cancer. Therefore, choose A for this question. |
| **GPT-4 Response** |
| Answer: A. Metoclopramide. Analysis: This question examines the contraindications of antiemetic drugs. Metoclopramide is a dopamine D2 receptor antagonist, and its anti-emetic effect is achieved by blocking central and peripheral 5-HT3 receptors and dopamine D2 receptors. Elderly patients over 65 years of age are contraindicated for metoclopramide, because the drug may increase central nervous system adverse reactions in the elderly population, such as drowsiness, ataxia, and akathisia. Other antiemetics, such as ondansetron, dexamethasone, diphenhydramine, and aprepitant, are safe in older patients. Therefore, the correct answer is A. |
| **ChatGPT Response** |
| Answer: B. Ondansetron. Analysis: Ondansetron is a 5-HT3 receptor antagonist, mainly used to prevent nausea and vomiting caused by chemotherapy for malignant tumors. However, ondansetron is contraindicated in patients with hepatic insufficiency or pregnancy, because these patients have reduced ability to metabolize and clear drugs, which may easily cause drug accumulation and cause side effects. For the specific situation of the patient, it should be selected according to the clinical judgment of the doctor. But in this case, ondansetron is contraindicated, so the correct answer is B. |

Table 1: A translated example from our ExplainCPE dataset with response of GPT-4 and ChatGPT (✓: correct answer option). The blue text represents the given answer in the response. The red text represents the error in the GPT-4 response, the reason for choosing the option is breast cancer rather than old age.

we find substantial limitations in understanding of these LLMs over medical text and their ability to execute computational reasoning effectively. For example, only GPT-4 passed Chinese Pharmacist Examination with 75.7% accuracy, while other models like ChatGPT failed. Through the case analysis of GPT-4 and ChatGPT, we found that the explanations generated by LLMs still have flaws such as contradictory, insufficient analysis, confused logic, and how to improve its interpretability is the part that LLMs should pay attention to in the future. Furthermore, we report heterogeneous preferences for in-context learning among different LLMs, suggesting varying strategies for explanation generation. For example, models with little chatting ability such as BELLE (Ji et al., 2023b,a) are more sensitive to the number of few-shot examples than with ChatGPT with strong chatting ability. To the best of our knowledge, we are the first to propose a free-text explanation benchmark in Chinese medical examination and further explore the interpretability of LLMs in the medical field. We provide a baseline for future research on explanation generation research, and this dataset can also be used to improve the interpretability of these large language models. As the broader issues of AI safety and trustworthiness gain attraction, our work represents a pioneering step towards enhancing the medical interpretability of LLMs, underscoring the urgent need to develop AI that is not only intelligent, but also transparent, robust, unbiased and reliable.

Our main contributions can be summarized as follows:

- We introduce ExplainCPE, a challenging benchmark for generating free-text explanations in Chinese medical QA, which provides a baseline for future research on explanation generated by LLMs, and can be used to study how to improve the ability of the model to generate explanation.

- We analyze the basic attributes of the dataset, such as the average length of questions, options, and explanations. Additionally, we examine the high-level categories of questions, which can assist researchers in understanding the distribution of categories in ExplainCPE and the interpretability performance of the models.

- We conduct experiments on the ExplainCPE dataset to demonstrate its effectiveness and feasibility. Our findings reveal that different LLMs exhibit varying preferences for in-context learning. We analyze error cases and identify some limitations of current LLMs, which can serve as directions for future development.

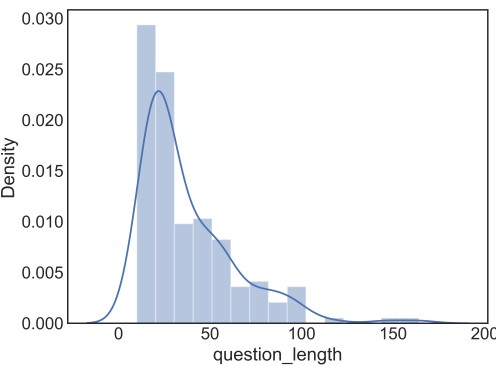 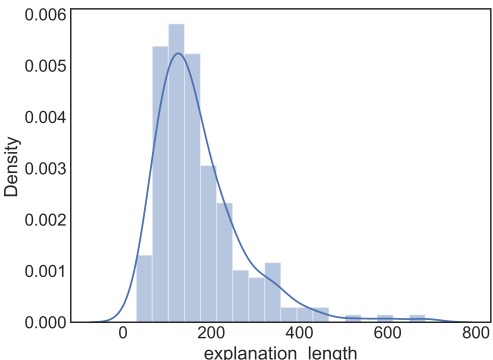

Figure 1: Distribution of questions and explanations length in ExplainCPE.

## 2 Related Work

### 2.1 Medical Question Answering

In the medical domain, addressing questions can be particularly challenging due to their specialized and complex nature. Consequently, community efforts have been directed towards advancing biomedical question-answering systems, such as BioASQ (Tsatsaronis et al., 2012, 2015). Another system, SeaReader (Zhang et al., 2018), was proposed to answer clinical medical questions by leveraging documents extracted from medical publications. In a study by Yue et al. (2020), the authors performed a comprehensive analysis of the emrQA (Pampari et al., 2018) dataset to evaluate the capacity of QA systems to utilize clinical domain knowledge and generalize to novel questions. Furthermore, Jin et al. (2019) introduced PubMedQA, a system that generates questions based on article titles and can be answered using their respective abstracts. Li et al. (2020) developed a large-scale medical multiple-choice question dataset and proposed a novel reading comprehension model, KMQA, capable of incorporating both structural medical knowledge and plain text.

### 2.2 Free-text Explanation

Since deep learning became the dominant paradigm in NLP research, how to interpret the predictions of neural models has become an essential part of model transparency. In explainable NLP, various forms of explanations exist, including extractive rationales, semi-structured, structured explanations, and free-text explanations. Saha et al. (2022) examine the impact of sample hardness on the capacity of both LLMs and humans to elucidate data labels. Camburu et al. (2018) augment

the SNLI dataset by introducing e-SNLI, which encompasses an additional layer of human-annotated natural language explanations for entailment relations. Rajani et al. (2019) gather human explanations for commonsense reasoning in the form of natural language sequences and highlighted annotations within a novel dataset known as Common Sense Explanations (CoS-E). Aggarwal et al. (2021) develop a first-of-its-kind dataset named ECQA, comprising human-annotated positive and negative properties, as well as free-flow explanations for 11,000 question-answer pairs derived from the CQA dataset. Ye and Durrett (2022) assess the performance of four LLMs across three textual reasoning datasets utilizing prompts containing explanations in multiple styles. Their findings indicate that human-evaluated high-quality explanations are more likely to coincide with accurate predictions.

### 2.3 LLMs Benchmarks

New NLP benchmarks are urgently needed to align with the rapid development of LLMs. MMLU(Hendrycks et al., 2021) is a collection of English-language materials that encompasses knowledge from 57 different disciplines including elementary mathematics, US history, computer science, law, and more. To attain high accuracy on this test, models must possess extensive world knowledge and problem solving ability. Another significant contribution to this field is the C-EVAL(Huang et al., 2023), which represents the first comprehensive effort to evaluate foundational models' knowledge and reasoning capabilities within a Chinese context. C-EVAL consists of multiple-choice questions designed to assess performance across four difficulty levels:

middle school, high school, college, and professional. These questions cover 52 diverse disciplines, spanning from humanities to science and engineering, thereby providing a holistic evaluation of the model's capabilities. Zhang et al. (2023) introduces the GAOKAO-Benchmark (GAOKAO-Bench), an intuitive benchmark that employs questions from the Chinese Gaokao examination as test samples for evaluating LLMs. Most benchmarks focus on evaluating the performance of LLMs in answering or answering questions, but few focus on the ability of LLMs to explain the answers given.

## 3 ExplainCPE Dataset

### 3.1 Dataset Collection

The National Licensed Pharmacist Examination in China, collaboratively administered by the Ministry of Personnel and the State Food and Drug Administration, serves as the basis for our question set.

In order to evaluate the performance and generalizability of our models, we have compiled a test set using examples from the previous two years (2020-2021) of the official examination. Each official question's explanation is sourced from official examination solution. Additionally, we have collected over 7,000 instances from various sources, including the internet and exercise books. The instance in ExplainCPE dataset is multiple choice question with five options.

In addition to the official questions, we also collaborated with three doctoral students from Peking Union Medical College (all of whom have undergone standardized residency training). They manually reviewed 320 samples from the collected data to evaluate the completeness and accuracy of the label and explanations. The evaluation resulted in the 99.4%/99.0% accuracy rate, with 318/317 out of the 320 samples being deemed correct.

Following the removal of duplicate and incomplete questions (e.g., those lacking answers or options), we randomly divided the remaining instances into training and development sets based on a predetermined ratio. To further enhance the quality of our dataset, we inspected instances with an edit distance of less than 0.1 and manually removed questions containing different words that conveyed the same meaning.

|  | Train | Dev | Test |
|---|---|---|---|
| #Questions | 6867 | 500 | 189 |
| Avg. words of Q | 28.31 | 28.44 | 37.79 |
| Avg. words of O | 8.12 | 8.55 | 9.76 |
| Avg. words of E | 120.52 | 116.32 | 171.94 |
| Max words of Q | 338 | 259 | 164 |
| Max words of O | 146 | 95 | 57 |
| Max words of E | 1011 | 604 | 685 |
| Options per problem | | 5 | |

Table 2: ExplainCPE dataset statistics, where Q, A, E represents the Question, Answer, and Explanation, respectively.

### 3.2 Data Statistic

The training, development, and test sets comprise 6,867, 500, and 189 questions, respectively, with average lengths of 28.31, 28.44, and 37.79 words. A summary of the dataset statistics can be found in Table 2. Figure 1 illustrates the distribution of question and explanation lengths across the training, development, and test sets.

### 3.3 Data Analysis

In order to investigate the properties of the ExplainCPE dataset, we primarily focus on the diversity of questions in this subsection. Our aim is to determine the categories of problems that LLMs excel at handling. To achieve this, we performed a multi-level classification of the dataset, comprising three levels.

At the first level, questions are classified into positive and negative categories. Positive questions which is also called direct question prompt the respondent to select the correct option, while negative questions require identifying the incorrect option among the options provided.

At the second level, questions are categorized into 7 groups: logical reasoning, drug knowledge, scenario analysis, mathematical calculation, disease knowledge, general knowledge, and others.

Finally, at the third level, questions are classified into 14 categories based on their content: anti-inflammatory, infection, tumor, anesthesia, cardiovascular, weight loss, orthopedics, nervous system, respiratory system, digestive system, urinary system, endocrine, immune system, and others.

We randomly selected 1200 instances from the training and development sets and manually assigned a three-level classification to each question. The proportional distribution of each category

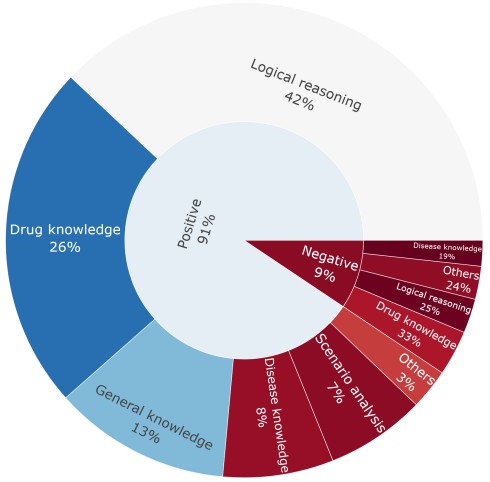

Figure 2: The distributions of proportions for each category at two levels in ExplainCPE. In the first layer of categories, positive questions account for the majority. In the second category, logic questions and knowledge questions account for the majority.

within the dataset is presented in Figure 2. A more detailed proportional distribution of each category within the dataset is presented in Appendix B.

## 4 Experiments

### 4.1 Prompting

Prompting has a significant impact on the output of generative language models, so we standardized the structure of our prompts. In order to better analyze the performance and interpretability of language models, we designed prompts to request the model to provide an answer option along with an explanation in the test set. An example of the template and a fully instantiated prompt can be found in Appendix A. Two types of prompt templates were utilized: with and without instructions. The purpose of this design was to explore the influence of instructions on different models. In the zero-shot setting, the few_shot_example slot was left blank. Additionally, it should be noted that prompts without instructions are the same as prompts with instructions in the zero-shot setting.

To investigate the impact of in-context on model performance, we designed prompts with different numbers of few-shot examples, including zero-shot, one-shot, four-shot, and eight-shot prompts. For one-shot prompts, we randomly selected a single instance from the training set. For four-shot and eight-shot prompts, we manually selected instances with varying question types to ensure model predictions were balanced. It should be noted that the

few-shot examples were the same for all models in each respective prompt type.

### 4.2 Model Comparison

To compare the performance of different models, we evaluated several LLMs on our test dataset. We recognize that LLMs can be classified as chat or non-chat models, depending on their ability to engage in human-like conversation. Chat models, which are pre-trained with vast amounts of data and fine-tuned through reinforcement learning from human feedback (RLHF), include GPT-4 (OpenAI, 2023), ChatGPT (OpenAI, 2022), ChatGLM-6B (Du et al., 2022; Zeng et al., 2023), and ChatYuan (ClueAI, 2023). Non-chat models, on the other hand, are typically pre-trained on unsupervised plain text and fine-tuned on code or instructional data but do not have sufficient RLHF to enable human-like conversation. Examples of non-chat models include GPT-3 (Ouyang et al., 2022), BELLE (Ji et al., 2023b,a), and GPT-3 (Brown et al., 2020). Consequently, non-chat models are more inclined to predict the next word or complete a given task rather than engage in conversation. In this section, we provide a brief introduction to the LLMs used in our experiments.

- ChatGPT (OpenAI, 2022) is a large language model with hundreds of billions of parameters, specifically designed for human-like conversation across a wide range of topics. ChatGPT's text understanding ability is derived from language model pre-training, its reasoning ability is derived from code pre-training, its logical reasoning ability is derived from supervised instruction training, and its dialogue ability is derived from RLHF.

- GPT-4 (OpenAI, 2023) represents the latest milestone in OpenAI's deep learning scaling efforts, and is a large multimodal model that exhibits human-level performance on various professional and academic benchmarks. GPT-4 outperforms ChatGPT on most tasks.

- GPT-3 (Ouyang et al., 2022) is a series of models. In this paper, we simply call text-davinci-003 with GPT-3. Text-davinci-003 is capable of performing any language task with better quality, longer output, and more consistent instruction-following than GPT-3.

- ChatGLM-6B (Du et al., 2022; Zeng et al.,

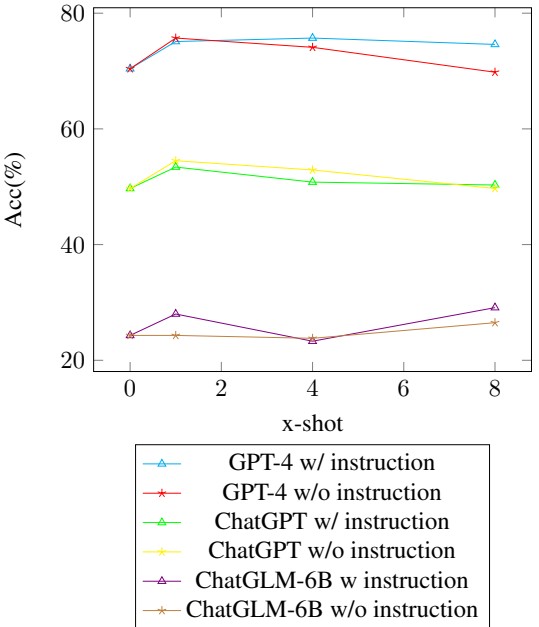 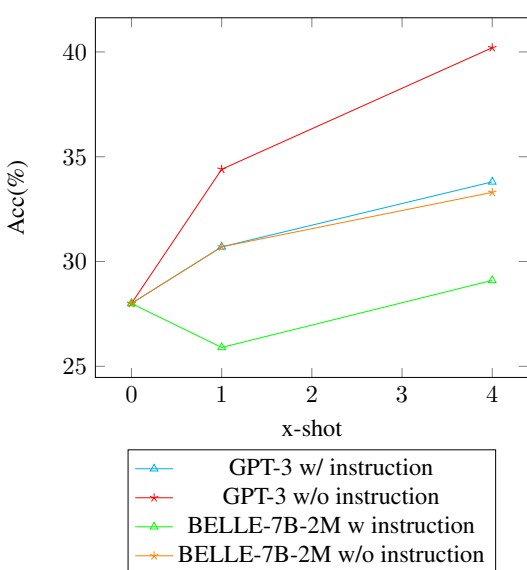

Figure 3: Performance comparison under different few-shot numbers. **Left**: Models with chatting ability such as GPT-4, ChatGPT and ChatGLM-6B. **Right**: Models without enough chatting ability such as GPT-3 and BELLE. Each model has 2 settings, with instruction and without instruction.

2023) is an open-source dialogue language model that supports both Chinese and English bilinguals. Utilizing technology similar to ChatGPT, it is optimized for Chinese question-answering and dialogue. After about 1T identifiers of Chinese and English bilingual training, supplemented by supervision, fine-tuning, feedback self-help, human feedback reinforcement learning, and other technologies, ChatGLM-6B with 6.2 billion parameters can generate answers that closely align with human preferences.

- The BELLE-7B-2M (Ji et al., 2023b,a) model is based on Bloomz-7b1-mt and trained on 2M pieces of Chinese data, combined with 50,000 pieces of English data open-sourced by Stanford-Alpaca. It has demonstrated good Chinese instruction understanding and response generation capabilities.

- ChatYuan (ClueAI, 2023) is further trained based on PromptCLUE-large (Zhang and Xu, 2022), combined with hundreds of millions of functional dialogues and multiple rounds of dialogue data. ChatYuan is capable of answering questions in fields such as law, and can be used for question-answering, dialogue in context, creative writing.

| Model | Acc(%) | Rouge-1 | Rouge-2 | Rouge-L |
|---|---|---|---|---|
| GPT-4 | **75.7** | **0.384** | **0.140** | **0.247** |
| ChatGPT | 54.5 | 0.341 | 0.114 | 0.216 |
| GPT-3 | 40.2 | - | - | - |
| ChatGLM-6B | 29.1 | 0.315 | 0.099 | 0.184 |
| BELLE-7B-2M | 33.3 | - | - | - |
| ChatYuan | 27.0 | - | - | - |

Table 3: Performance comparison on ExplainCPE dataset.

## 5 Results

One of the main objectives of our dataset is to evaluate the interpretability of models by assessing the quality of the generated text. Therefore, we not only measured the accuracy of the models but also required a useful and efficient evaluation metric. Evaluating interpretability is a long-standing problem due to the diversity of interpretation forms and content.

As suggested by Wiegreffe et al. (2022), the quality of explanations generated by the joint method needs to be further verified. Therefore, we chose two methods to evaluate the interpretability of the models: automatic metrics and human evaluation. For automatic metrics, we used Rouge to measure the quality of the explanations provided by the models and accuracy to measure the models' performance. Due to the model's input length limitation, we could not conduct eight-shot experi-

|  |  | Well-formed | Support | Correctness | Validity | Novelty |
|---|---|---|---|---|---|---|
| **Human Evaluation** | GPT-4 | 4.99 | 4.96 | 4.03 | 4.60 | 4.57 |
| | ChatGPT | 4.99 | 4.96 | 3.16 | 4.24 | 4.39 |
| | ChatGLM-6B | 4.48 | 3.93 | 2.05 | 3.16 | 3.37 |
| **GPT-4 Evaluation** | GPT-4 | 3.72 | 4.54 | 4.42 | 4.30 | 4.10 |
| | ChatGPT | 3.42 | 4.00 | 3.84 | 3.94 | 3.84 |
| | ChatGLM-6B | 2.90 | 3.32 | 3.06 | 3.24 | 3.34 |

Table 4: Human evaluation results(top) and GPT-4 evaluation results(bottom) of different models in five perspectives.

ments for some models, such as GPT-3. Moreover, some models did not respond with answers and explanations even when we requested them, which is why some models lack a Rouge score.

## 5.1 Automatic Metrics

The performance of each model in each setting can be found in Appendix C. Table 3 presents the best performance of each model on the test set, regardless of whether the prompt of each model is consistent. Not surprisingly, GPT-4 is the best-performing model, achieving 75.7% accuracy with the most suitable set of 1-shot without instruction. Therefore, GPT-4 has demonstrated the ability to pass the National Licensed Pharmacist Examination in China and has outperformed more than 80% of the people who take the examination. In contrast, ChatGPT achieved an accuracy rate of 54.5% on the test set, which would not be sufficient to pass the examination. GPT-3, ChatGLM-6B, BELLE-7B-2M, and ChatYuan achieved accuracies of 40.2%, 29.1%, 33.3%, and 27.0% on the test set, respectively. Models with fewer parameters generally perform worse than models with more parameters. This can be attributed to smaller parameter sizes, which means that models may not have the capacity to remember enough knowledge or understand the meaning of context.

Regarding interpretable automatic evaluation indicators, GPT-4 achieved the best results in explanation generation with a Rouge-L score of 0.247, followed by ChatGPT, ChatGLM-6B, and BELLE-7B-2M. ChatGLM-6B yielded unexpected results in metrics, despite its relatively small parameter size, with high accuracy and Rouge scores.

We plotted line charts of model performance as a function of the number of few-shots. The line chart is divided into two, the chart on the left for chat models and the chart on the right for non-chat models. From the figure, we identified three key findings.

Firstly, it is evident from Figure 3 that regard-less of the size of the model parameters or whether instructions are given, going from zero-shot to one-shot often results in a significant performance improvement, which is better than any subsequent increase of few-shot examples.

Secondly, when comparing chat models and non-chat models, GPT-3 is a model with a large number of parameters but weak dialogue ability, while GPT-4 and ChatGPT are models with strong dialogue ability. Regardless of whether instructions are provided, the performance of GPT-3 increases with an increase in few-shot examples, but GPT-4 and ChatGPT tend to achieve their maximum performance in one-shot setting. This suggests that for a model with a large number of parameters and strong dialogue ability, one-shot setting is a good choice. Conversely, for models with weak dialogue ability, their performance is somewhat proportional to the number of few-shot examples.

Thirdly, when comparing the two figures, the models in the left picture have strong dialogue ability. Therefore, in the case of the same number of few-shot examples, providing instructions is better than not providing instructions. However, in the right picture, the models have weak dialogue ability. Therefore, in the case of the same number of few-shot examples, not providing instructions is better.

## 5.2 Human Evaluation

From the perspective of interpretability, there are certain limitations in using the rouge evaluation metric to evaluate the interpretability of the model. So we also used human evaluation to assess the qualitative properties of the generated explanations. We follow Monsen and Rennes (2022), Wiegreffe et al. (2022) and Kunz et al. (2022) asking annotators to rate from 1 to 5 according to the following questions for each $e$.

- Is $e$ a well-formed sentence?

- Does $e$ support the label?

| Type | All | Positive | Negative | Logical | Drug | Scenario | Math | Disease | General | others |
|---|---|---|---|---|---|---|---|---|---|---|
| Num of type | 189 | 128 | 61 | 78 | 45 | 38 | 2 | 3 | 17 | 6 |
| GPT-4 | 75.7 | 74.2 | 78.7↑ | 76.9↑ | 73.3 | 76.3↑ | 100.0↑ | 100.0↑ | 70.6 | 66.7 |
| ChatGPT | 54.5 | 52.3 | 59.0↑ | 51.3 | 60.0↑ | 47.4 | 00.0 | 100.0↑ | 58.8↑ | 83.3↑ |
| GPT-3 | 40.2 | 40.6↑ | 39.3 | 38.5 | 46.7↑ | 34.2 | 00.0 | 0.33 | 58.8↑ | 16.7 |
| ChatGLM-6B | 29.1 | 31.3 | 24.6 | 26.9 | 31.1 | 31.6 | 50.0↑ | 66.7↑ | 17.6 | 33.3 |
| BELLE-7B-2M | 33.3 | 40.6↑ | 18.0 | 39.7↑ | 31.1 | 21.1 | 50.0↑ | 33.3 | 41.2↑ | 16.7 |
| ChatYuan | 27.0 | 28.9↑ | 23.0 | 25.6 | 24.4 | 26.3 | 50.0↑ | 33.3↑ | 41.2↑ | 16.7 |

Table 5: Performance of models on different types of samples in ExplainCPE.

- Is the content of *e* factually correct?

- Does *e* provide a valid reasoning path for the label?

- Does *e* add new information, rather than recombining information from the input?

Due to the low accuracy of some models and the poor quality of the generated explanations, we only manually evaluated a subset of the models, and the results are presented in Table 4. As expected, GPT-4 remains the best performer. It is noteworthy that the performance of GPT-4 and ChatGPT in terms of well-formed and support is the same. This indicates that both GPT-4 and ChatGPT can comprehend the question's requirements, provide the label, and generate a complete and coherent explanation that supports the label. However, GPT-4 outperforms ChatGPT in terms of the correctness of the explanation, effectiveness of the explanation process, and novelty. ChatGLM lags behind ChatGPT and GPT-4 on all five indicators. And We also ask GPT-4 to evaluate responses, the relative scores of different metrics are consistent with human evaluation results.

### 5.3 Error Analyses

In this subsection, we present an analysis of the performance of the models from an overall perspective and specific examples. Table 5 displays the performance of the models on different types of questions. Notably, GPT-4 and ChatGPT perform better on negative questions, while other models perform better on positive questions. Moreover, GPT-4 demonstrates improvement in logical reasoning, whereas other models do not. While GPT-4 improves in scenario analysis questions, other models exhibit a decline. Conversely, GPT-4 declines in general knowledge questions while other models improve. GPT-4 correctly solves two mathematical calculation questions, whereas ChatGPT

fails on all such questions. These findings suggest that GPT-4 has stronger logical reasoning, scenario analysis, and mathematical calculation abilities than other models. The superior performance of GPT-4 and ChatGPT on negative questions indicates their better understanding of text and ability to answer questions.

We analyze specific error cases of ChatGPT and GPT-4 to identify the limitations of current LLMs. Appendix D outlines the reasons for explanation errors. Although the results of LLMs are impressive, they are not yet perfect. In example 1, GPT-4 provides the correct answer but the wrong explanation, which is difficult to detect. Thus, models should pay close attention to such errors before widespread use. In Example 2, although the model has a certain calculation capability, the reliability of its calculation is still not guaranteed. In Example 3, neither GPT-4 nor ChatGPT fully comprehends the detailed requirements of the question, leading to errors. Therefore, LLMs still have scope for improvement in text comprehension and generating explanations.

## 6 Conclusion

In this work, we propose ExplainCPE, a challenging medical dataset for natural language explanation evaluation. Our study on ExplainCPE dataset demonstrates the potential of LLMs in medical question answering with explanations. Our analysis of model performance on different types of questions reveals the strengths and limitations of different LLMs in terms of in-context learning. The error cases point out the need for further improvement in LLMs in explanation generation and text comprehension. Further work can use our dataset to improve and evaluate the model interpretability.

## Limitations

Due to the lack of interpretable benchmarks in the medical professional field, we present ExplainCPE in this paper. While there are many explainable methods, we only contribute to the Explanation Generation. Moreover, most of the current interpretable methods are aimed at classification tasks. For LLMs which are used to generate response, new interpretable methods are necessary. We explore the ability of LLMs in medical diagnosis and interpretability. While model performance can be well assessed by accuracy, automatic assessment of interpretability is still lacking.

However, our analysis of ExplainCPE dataset is just a preliminary exploration, and there is still much room for further research and development. For example, future work can focus on improving the quality and diversity of the explanations in the dataset, expanding the coverage of medical knowledge, and exploring new evaluation metrics for interpretability. In addition, more advanced LLMs can be developed to further improve the performance of medical question answering with explanations by utilizing the data in the training set. We believe that the ExplainCPE dataset can serve as a valuable resource for the research community to advance the field of medical question answering and LLMs.

## Ethical Statement

This paper is concerned about proposing a dataset on explanations of medical question answers. The data in this dataset are all from Chinese Pharmacist Examination related exercises. Moreover, the cases in the exercises are all fictitious cases, and there is no personal privacy, discrimination or attack content. Judging from its impact, this data set can be used to improve the interpretation ability in human medical diagnosis, reduce misdiagnosis, and contribute to human intelligent medical care.

## Acknowledgments

We thank Yuxiang Wu for valuable feedback and support on running experiments. We thank Xinshuo Hu for helpful discussions and suggestions. This work is jointly supported by grants: Natural Science Foundation of China (No. 62006061, 82171475), Strategic Emerging Industry Development Special Funds of Shenzhen (No.JCYJ20200109113403826).

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

## A Prompting Template

There are two types of prompt templates, prompting with instruction and prompting without instruction. And you can check the template in Table 6. You can also see the template instantiation in Table 7.

## B Distribution of Categories

In Figure 4, we show the proportion distribution of each type in the dataset in more detail.

## C Performance Comparison

Perhaps due to the training data or model size, these models do not respond well to a given multiple-choice question. We have already evaluated two popular medical LLMs—ChatGLM-Med and Huatuo-Llama-Med-Chinese—and found that they struggled with our multi-choice questions.

Wang et al. (2023b) constructed a Chinese medical instruction data set through the medical knowledge graph and GPT3.5 API, and based on this, fine-tuned the instructions of ChatGLM-6B to improve the question-answering effect of ChatGLM in the medical field called Med-ChatGLM (Wang et al., 2023b). Based on the same data, we also trained a medical version of the LLaMA model called Huatuo (Wang et al., 2023a).

In Table 8, we show the results of each model under all settings, including the number of few-shots and with or without instructions.

## D Error Example

In this section, we mainly show some classic errors made by ChatGPT and GPT-4. We analyzed these errors in order to study the current shortcomings of LLMs.

**Example 1**
**Question**:The patient, female, 65 years old, received the second cycle of chemotherapy for breast cancer. Which antiemetic drugs are prohibited?
**Option A**:Metoclopramide.✓
**Option B**:Ondansetron.
**Option C**:Dexamethasone.
**Option D**:Diphenhydramine.
**Option E**:Aprepitant.
**Gold Explanation**: This question examines antiemetic drugs. Optional antiemetic drugs are mainly dopamine receptor blockers (such

as metoclopramide), 5-HT3 receptor blockers (such as ondansetron, tropisetron, and palonosetron, etc.) , corticosteroids (such as dexamethasone), anticholinergics and antihistamines (such as diphenhydramine), and NK-1 receptor blockers (such as aprepitant), etc. Special emphasis should be placed on the prohibition of metoclopramide as an antiemetic in patients with breast cancer. Therefore, choose A for this question.
**ChatGPT Response**: Answer: B. Ondansetron. Analysis: Ondansetron is a 5-HT3 receptor antagonist, mainly used to prevent nausea and vomiting caused by chemotherapy for malignant tumors. However, ondansetron is contraindicated in patients with hepatic insufficiency or pregnancy, because these patients have reduced ability to metabolize and clear drugs, which may easily cause drug accumulation and cause side effects. For the specific situation of the patient, it should be selected according to the clinical judgment of the doctor. But in this case, ondansetron is contraindicated, so the correct answer is B.
**GPT-4 Response**: Answer: A. Metoclopramide. Analysis: This question examines the contraindications of antiemetic drugs. Metoclopramide is a dopamine D2 receptor antagonist, and its anti-emetic effect is achieved by blocking central and peripheral 5-HT3 receptors and dopamine D2 receptors. Elderly patients over 65 years of age are contraindicated for metoclopramide, because the drug may increase central nervous system adverse reactions in the elderly population, such as drowsiness, ataxia, and akathisia. Other antiemetics, such as ondansetron, dexamethasone, diphenhydramine, and aprepitant, are safe in older patients. Therefore, the correct answer is A.
**Error Analyse**: GPT-4 is right. ChatGPT is wrong. Although GPT-4 makes a right prediction, its explanation is flawed because the reason given by GPT-4 is old age but the real reason is cancer.

**Example 2**
**Question**: 100 mg of a certain drug is injected intravenously, blood samples are collected immediately, and the drug concentration is measured to be 5 $\mu$ g/ml, what is the apparent volume of distribution?
**Option A**: 5L.

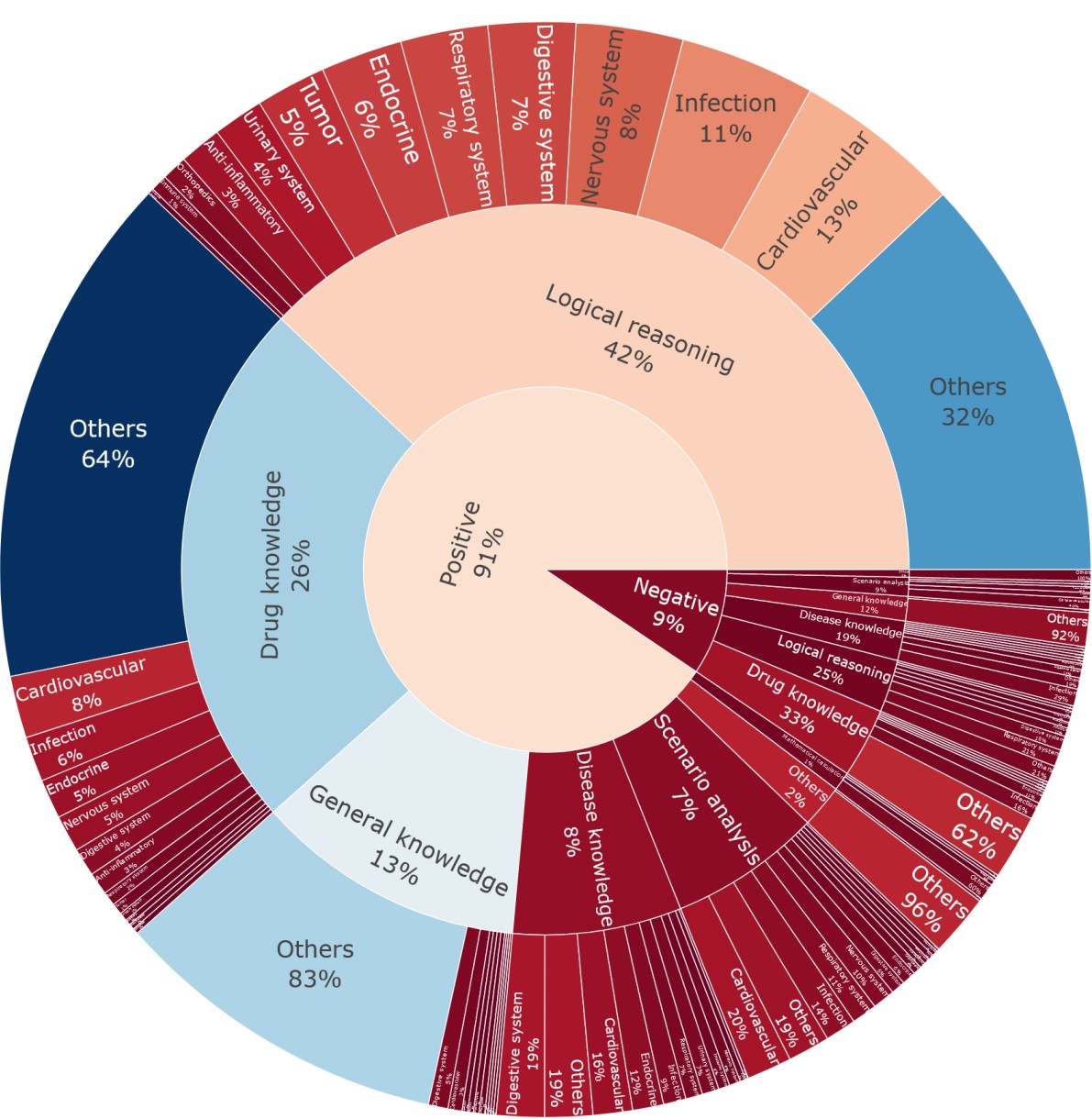

Figure 4: Proportional Distribution of each category of ExplainCPE dataset.

| Type | Prompt Template |
|------|-----------------|
| w/ instruction | 该题为单选题,请你回答下列问题并给出解析。
{{few_shot_examples}}
问题：{{question}}
{{options}} |
| w/ instruction | This question is a multiple-choice question, please answer the following questions and give an explanation.
{{few_shot_examples}}
Question：{{question}}
{{options}} |
| w/o instruction | {{few_shot_examples}}
问题：{{question}}
{{options}} |
| w/o instruction | {{few_shot_examples}}
Question：{{question}}
{{options}} |

Table 6: Prompt template. The solid line separates whether it is with instruction. And the dotted line separates the Chinese and English versions.

---

该题为单选题,请你回答下列问题并给出解析。
问题：药物的光敏性是指药物被光降解的敏感程度。下列药物中光敏性最强的是?
选项A：氯丙嗪
选项B：硝普钠
选项C：维生素B2
选项D：叶酸
选项E：氢化可的松

This question is a multiple-choice question, please answer the following questions and give an explanation.
Question: The photosensitivity of a drug refers to the sensitivity of the drug to photodegradation. Which of the following drugs is the most photosensitizing?
Option A: Chlorpromazine.
Option B: Sodium nitroprusside.
Option C: Vitamin B2.
Option D: Folic acid.
Option E: Hydrocortisone.

Table 7: Instantiated example of prompt template. The dotted line separates the Chinese and English versions.

---

**Option B**: 2L.
**Option C**: 20L.✓
**Option D**: 50L.
**Option E**: 29ml.
**Gold Explanation**: This question examines the calculation of the apparent volume of distribution. The formula for calculating the apparent volume of distribution is "V=X/C", where V is the apparent volume of distribution, X is the amount of drug in the body, and C is the blood drug concentration. V=X/C=100mg/5$\mu$g/ml=20L. Others are interference options, so choose C for this question.

**ChatGPT Response**: Answer: D.50L. Analysis: Apparent volume of distribution (Vd) is a parameter representing the distribution range of a drug, and its calculation formula is: Vd = total drug dose/drug concentration. In this question, the total dose of the drug is 100 mg, and the drug concentration is 5 $\mu$g/ml (note that the unit should be the same). The total dose of the drug is converted to $\mu$g to be 100,000 $\mu$g, which is substituted into the calculation formula: Vd = 100,000 $\mu$g/5 $\mu$g/ml = 20,000ml = 20L. Because the distribution of drugs not only occurs in the blood, but may also be distributed in cells, tissues and organs, etc., the apparent distribution volume of common drugs is relatively large, generally ranging from tens of liters to hundreds of liters (ie: 0.03L/kg to 0.8L/kg). The answer to this question is option D, and 50L is within the apparent volume of distribution of common drugs.

**GPT-4 Response**: Answer: C.20L. Explanation: The apparent volume of distribution (Vd)

is a theoretical value used to describe the range of drug distribution in the body. The formula for calculating Vd is: Vd = drug dose / initial drug concentration. In this question, the drug dose is 100 mg, and the measured drug concentration is 5 $\mu$g/ml. First convert the dose to micrograms: 100mg = 100,000 $\mu$g. Then apply the Vd formula: Vd = 100,000 $\mu$g / 5 $\mu$g/ml = 20,000ml, which is 20L. Therefore, the apparent volume of distribution of the drug is 20L, and the correct answer is C.

**Error Analyse**: GPT-4 is right. ChatGPT is wrong. ChatGPT gets an answer in the middle process, but it do not take that answer.

**Example 3**

**Question**: What are the medicines used to prevent relapse and transmission and to prevent malaria?

**Option A**: Quinine.
**Option B**: Hydroxychloroquine.
**Option C**: Primaquine.✓
**Option D**: Amodiaquine.
**Option E**: Artemisinin.

**Gold Explanation**: This question examines the action characteristics of primaquine. Primaquine can kill Plasmodium vivax, Malaria malaria, and Plasmodium falciparum. It can also kill the gametocytes of various Plasmodium parasites, and is clinically used as the drug of choice for controlling recurrence and preventing transmission. Quinine, chloroquine, and artemisinin are used as drugs to control the symptoms of malaria. So choose C for this question.

**ChatGPT Response**: Answer: E. Artemisinin. Analysis: Artemisinin is currently an effective drug for treating malaria, especially for falciparum malaria and refractory malaria. In addition, artemisinin also has certain antitumor and antiviral effects. Quinine is another drug used to treat malaria, but it is mainly used to treat the symptoms and complications of malaria, not to treat malaria itself directly. Hydroxychloroquine, primaquine, and amodiaquine are also drugs used to treat malaria, but drug resistance has emerged and their use is limited. Therefore, artemisinin is considered to be the drug of choice for the treatment of malaria.

**GPT-4 Response**: Answer: B. Hydroxychloroquine. Analysis: This question examines malaria prevention drugs. Hydroxychloro-

quine is an antimalarial drug used for the prevention and treatment of malaria, especially against infection by susceptible strains of Plasmodium. It can not only kill the malaria parasite, but also reduce the transmission probability of the malaria parasite, so it is used to prevent the recurrence and transmission of malaria. Other options Quinine, primaquine, amodiaquine, and artemisinin are mainly used in the treatment of malaria. So the correct answer is B.

**Error Analyse**: Both GPT-4 and ChatGPT are wrong. The question asks for a drug to prevent malaria, but ChatGPT gives a drug to treat malaria. GPT-4 says that Hydroxychloroquine can prevent malaria, but Hydroxychloroquine is actually used to control malaria, not to prevent malaria.

**Example 4**

**Question**: Which medicines require water restriction while taking?

**Option A**: cough syrup.✓
**Option B**: Diprophylline.
**Option C**: ursodeoxycholic acid.
**Option D**: ritonavir.
**Option E**: Alendronate Sodium.

**Gold Explanation**: This question examines drugs that should restrict drinking water. Cough medicines such as cough syrup, licorice mixture, etc. These viscous medicines will adhere to the inflamed throat and play a role. After use, drink less water, especially hot water, to avoid washing the local medicines. Taking theophylline or theophylline controlled-release tablets, aminophylline, cholophylline, dihydroxyprophylline, etc., because it can increase renal blood flow, has a diuretic effect, increases urine output and easily causes dehydration, dry mouth, Polyuria or heart palpitations; at the same time, asthmatics are often accompanied by low blood volume. Therefore, it is advisable to pay attention to the right amount of fluids and drink plenty of boiled water. Phenylpropanol, hydroxymethylcoumarol, dehydrocholic acid and ursodeoxycholic acid in choleretic drugs can cause excessive secretion of bile and diarrhea. Therefore, drink plenty of water when taking it to avoid dehydration caused by excessive diarrhea. Among protease inhibitors, ritonavir, indinavir, nelfinavir, amprenavir, lopinavir, etc., most of them can

cause urethral stones or kidney stones. Therefore, sufficient hydration should be ensured during the treatment period. In order to avoid the occurrence of stones, it is advisable to increase the daily water intake, and the daily drinking water should be more than 2000ml. Bisphosphonates are irritating to the esophagus and must be taken with more than 200ml of water; among them, sodium alendronate, sodium patradronate, and disodium clodronate can cause water when they are used to treat hypercalcemia. , Electrolyte disorder, so attention should be paid to replenishing fluids so that the daily urine output can reach more than 2000ml. Therefore, choose A for this question.

**ChatGPT Response**: Answer: B. Diprophylline. Analysis: Diprophylline is metabolized in the body to produce uric acid, and uric acid is easy to cause stones, so you need to limit the amount of water you drink when taking it. Drugs such as cough syrup, ursodeoxycholic acid, ritonavir, and alendronate do not require water restriction.

**GPT-4 Response**: Answer: E. Alendronate Sodium. Analysis: This question examines the contraindications and precautions of the drug. Alendronate Sodium (Alendronate Sodium) is a drug used to treat fractures, osteoporosis and other symptoms. It improves bone density by inhibiting bone resorption. Drinking water should be limited when taking alendronate sodium, because it needs to be taken on an empty stomach, and you must stand or sit upright for at least 30 minutes immediately, and avoid lying down after taking the medicine to prevent side effects such as esophagitis, stricture and ulcer. Other options, such as cough syrup, dyphylline, ursodeoxycholic acid, and ritonavir, do not require fluid restriction. Therefore, the correct answer is E. alendronate.

**Error Analyse**: Both GPT-4 and ChatGPT are wrong. The reason for the error is the lack of mechanism by which cough syrup works.

**Example 5**

**Question**:A 28-year-old female patient was given isotretinoin capsules because of acne. Which statement about drug education for this patient is false?

**Option A**:Contraceptive measures should be taken during treatment and within 3 months after drug withdrawal.

**Option B**:If skin and mucous membrane dryness occurs during treatment, the dose should be reduced or the drug should be discontinued.✓

**Option C**:Adverse reactions are common, but most of them can be recovered after stopping the drug.

**Option D**:The first 2 months of treatment may lead to abnormal liver enzymes, check liver function if necessary.

**Option E**:Option E: Some patients experienced transient exacerbation of rash after 2-4 weeks of medication.

**Gold Explanation**: This question examines the precautions for oral retinoic acid drugs. (1) Adverse reactions of isotretinoin are common, but most of them can be recovered after stopping the drug, and serious adverse reactions are rare or even rare. (2) The most common adverse reaction is dry skin and mucous membranes, and it is recommended to be used with skin barrier repair agents. (3) Rare adverse reactions include musculoskeletal pain, elevated blood lipids, abnormal liver enzymes, and dry eyes, which usually occur in the first 2 months of treatment, and should be used with caution in patients with obesity, dyslipidemia, and liver disease. (4) Isotretinoin has a clear teratogenic effect. Female patients of childbearing age and their spouses should strictly avoid pregnancy 1 month before treatment, during treatment and within 3 months after treatment. (5) It should be used with caution in patients who already have obvious depressive symptoms or suffer from depression. (6) Some patients will experience short-term exacerbation of skin rash after 2 to 4 weeks of use, which is usually transient, and those with severe reactions need to reduce the dose or even stop the drug. Therefore, choose B for this question.

**ChatGPT Response**: Answer: A. Contraceptive measures should be taken during treatment and within 3 months after drug withdrawal. Analysis: Isotretinoin capsule is a medicine used to treat acne, and its main ingredient is retinoic acid. Contraceptive measures should be taken during drug treatment and within 3 months after stopping the drug, because the drug can affect fetal development. Know the contraindications before treatment, such as pregnant women, lactating women, and those

with severe liver and kidney impairment. If dry skin and mucous membranes are found during treatment, the dose should be reduced or the drug should be stopped to avoid aggravating the condition. The first 2 months of treatment may lead to abnormal liver enzymes, so the patient's liver function should be checked before treatment, and if necessary, liver function should be checked regularly during the medication. Some patients experience transient exacerbation of the rash after 2 to 4 weeks of medication. This is a normal phenomenon, and there is no need to stop the medication, and it will be relieved within a few days. Therefore, the correct answer to this question is A.

**GPT-4 Response**: Answer: B. If skin and mucous membrane dryness occurs during treatment, the dose should be reduced or the drug should be discontinued. Analysis: Isotretinoin capsules may cause some adverse reactions during the treatment of acne. Contraceptive measures should be taken during treatment and within 3 months after drug withdrawal, because isotretinoin has a strong teratogenic effect (choice A is correct). Dry skin or dry mucous membranes may occur during treatment, but dose reduction or drug withdrawal is not required; usually it can be relieved by using mild skin care products, lip balm, artificial tears, etc. (option B is wrong). Adverse reactions are common, but most of them can be recovered after stopping the drug (choice C is correct). The first 2 months of treatment may lead to abnormal liver enzymes, check liver function if necessary (option D is correct). Some patients experienced transient exacerbation of rash during 2 to 4 weeks of medication (choice E is correct). Therefore, the wrong statement is option B.

**Error Analyse**: GPT-4 is right. ChatGPT is wrong. The explanation given by ChatGPT is correct and conforms to the gold explanation, but the label it gives is wrong.

**Example 6**

**Question**:The patient, male, 62 years old, developed bradykinesia 4 years ago and was diagnosed with Parkinson's disease. Symptoms are currently progressively worsening with mild cognitive impairment. Which drug should be used?

**Option A**:pramipexole.

**Option B**:Selegiline.
**Option C**:Adamantadine.
**Option D**:Entacapone.
**Option E**:Benserazide-levodopa.✓

**Gold Explanation**: This question examines the drug treatment of Parkinson's. 1. Early-onset patients without mental decline: (1) Pramipexole, selegiline or compound levodopa + COMT inhibitor (entecapone) should be the first choice if the treatment guidelines in the United States and Europe are followed; (2) If the patient cannot afford high-priced drugs due to economic reasons, amantadine is the first choice; (3) If due to the needs of special work, the patient strives to significantly improve motor symptoms or suffers from cognitive decline, the first choice is the first choice Compound levodopa or compound levodopa + COMT inhibitor (entecapone); (4) obvious tremor but poor efficacy of other drugs → trihexyphenidyl. 2. Late onset or accompanied by mental decline (1) Compound levodopa is generally the first choice for treatment. (2) Try not to use anticholinergic drugs, especially for elderly male patients. Therefore, choose E for this question.

**ChatGPT Response**: Answer: B. Selegiline Analysis: The main manifestation of Parkinson's disease is movement disorder, because Parkinson's disease is caused by the loss of dopaminergic neurons, so dopamine receptor agonists are one of the treatments for Parkinson's disease drugs. Pramipexole is also an effective treatment for Parkinson's disease in young people, but it is less suitable for this patient because of its more side effects in older patients. Amantadine works similarly to dopamine in treating Parkinson's disease, but it often causes side effects in elderly patients, so it is not suitable for this patient. Entacapone is a combination drug for the treatment of Parkinson's disease, which consists of levodopa and entapazide, but this drug is prone to side effects such as lowering of blood pressure, and is not suitable for elderly patients with mild cognitive impairment. Therefore, it is a better choice to choose B. selegiline, which is a selective dopamine D2 and D3 receptor agonist, which will not cause aggravation of dyskinesia symptoms such as muscle stiffness and tremor, and can improve cognitive cognitive barriers.

**GPT-4 Response**: Answer: D. Entacapone

Analysis: Parkinson's disease treatment drugs include dopamine precursors (levodopa and carbidopa), dopamine receptor agonists (selegiline, pramipexole), COMT inhibitors Entacapone, MAO-B inhibitors (Utahsi, Ruizuosi, etc.), anticholinergic amantadine. This patient is 62 years old, with a medical history of 4 years. The treatment effect of dopamine precursor combined with dopamine receptor agonist is more accurate. Considering the patient has mild cognitive impairment, the anticholinergic drug amantadine should be avoided. It is planned to use entacapone as the COMT inhibitor, which is beneficial to prolong the effect of dopamine prodrugs and improve the symptoms of patients. Therefore, Entacapone was chosen. The correct answer is D.

**Error Analyse**: Both are wrong. The model is more inclined to prescribe medicine to the disease, and it is difficult to make a diagnosis based on the specific situation of the patient.

| Model | Type | Prompt | Acc(%) | Rouge-1 | Rouge-2 | Rouge-L |
|---|---|---|---|---|---|---|
| GPT-4 | 0-shot | w/ instruction | 70.4 | 0.343 | 0.107 | 0.209 |
| GPT-4 | 1-shot | w/ instruction | 75.1 | 0.383 | 0.135 | **0.247** |
| GPT-4 | 4-shot | w/ instruction | **75.7** | 0.382 | 0.136 | 0.245 |
| GPT-4 | 8-shot | w/ instruction | 74.6 | 0.368 | 0.127 | 0.228 |
| GPT-4 | 0-shot | w/o instruction | 70.4 | 0.343 | 0.107 | 0.209 |
| GPT-4 | 1-shot | w/o instruction | **75.7** | **0.384** | **0.140** | **0.247** |
| GPT-4 | 4-shot | w/o instruction | 74.1 | 0.381 | 0.131 | 0.243 |
| GPT-4 | 8-shot | w/o instruction | 69.8 | 0.360 | 0.123 | 0.228 |
| ChatGPT | 0-shot | w/ instruction | 49.7 | 0.336 | 0.105 | 0.206 |
| ChatGPT | 1-shot | w/ instruction | 53.4 | 0.342 | 0.113 | 0.214 |
| ChatGPT | 4-shot | w/ instruction | 50.8 | 0.356 | 0.123 | 0.228 |
| ChatGPT | 8-shot | w/ instruction | 50.3 | 0.373 | 0.129 | 0.230 |
| ChatGPT | 0-shot | w/o instruction | 49.7 | 0.336 | 0.105 | 0.206 |
| ChatGPT | 1-shot | w/o instruction | 54.5 | 0.341 | 0.114 | 0.216 |
| ChatGPT | 4-shot | w/o instruction | 52.9 | 0.355 | 0.119 | 0.219 |
| ChatGPT | 8-shot | w/o instruction | 49.7 | 0.356 | 0.120 | 0.221 |
| GPT-3 | 0-shot | w/ instruction | 28.0 | - | - | - |
| GPT-3 | 1-shot | w/ instruction | 30.7 | - | - | - |
| GPT-3 | 4-shot | w/ instruction | 33.8 | - | - | - |
| GPT-3 | 0-shot | w/o instruction | 28.0 | - | - | - |
| GPT-3 | 1-shot | w/o instruction | 34.4 | - | - | - |
| GPT-3 | 4-shot | w/o instruction | 40.2 | - | - | - |
| ChatGLM-6B | 0-shot | w/ instruction | 24.3 | 0.281 | 0.082 | 0.156 |
| ChatGLM-6B | 1-shot | w/ instruction | 28.0 | 0.294 | 0.089 | 0.178 |
| ChatGLM-6B | 4-shot | w/ instruction | 23.3 | 0.310 | 0.094 | 0.184 |
| ChatGLM-6B | 8-shot | w/ instruction | 29.1 | 0.315 | 0.099 | 0.184 |
| ChatGLM-6B | 0-shot | w/o instruction | 24.3 | 0.281 | 0.082 | 0.156 |
| ChatGLM-6B | 1-shot | w/o instruction | 24.3 | 0.306 | 0.094 | 0.180 |
| ChatGLM-6B | 4-shot | w/o instruction | 23.8 | 0.309 | 0.097 | 0.185 |
| ChatGLM-6B | 8-shot | w/o instruction | 26.5 | 0.309 | 0.097 | 0.181 |
| BELLE-7B-2M | 0-shot | w/ instruction | 28.0 | 0.267 | 0.067 | 0.169 |
| BELLE-7B-2M | 1-shot | w/ instruction | 25.9 | 0.208 | 0.054 | 0.132 |
| BELLE-7B-2M | 4-shot | w/ instruction | 29.1 | - | - | - |
| BELLE-7B-2M | 0-shot | w/o instruction | 28.0 | 0.267 | 0.067 | 0.169 |
| BELLE-7B-2M | 1-shot | w/o instruction | 30.7 | 0.208 | 0.054 | 0.132 |
| BELLE-7B-2M | 4-shot | w/o instruction | 33.3 | - | - | - |
| ChatYuan | 0-shot | w/ instruction | 27.0 | - | - | - |
| ChatYuan | 1-shot | w/ instruction | 26.5 | - | - | - |
| ChatGLM-Med | 0-shot | w/ instruction | 17.5 | - | - | - |
| ChatGLM-Med | 1-shot | w/ instruction | 10.1 | - | - | - |
| Huatuo-Llama-Med-Chinese | 0-shot | w/ instruction | 18.5 | - | - | - |
| Huatuo-Llama-Med-Chinese | 1-shot | w/ instruction | 19.6 | - | - | - |

Table 8: Performance comparison on ExplainCPE.