# OpenReview forum: "ExplainCPE: A Free-text Explanation Benchmark of Chinese Pharmacist Examination"
_EMNLP/2023/Conference — EMNLP 2023 Findings_

### Official Review · Reviewer_CyCb · 2023-08-01

**Soundness:** 3

**Excitement:**

3: Ambivalent: It has merits (e.g., it reports state-of-the-art results, the idea is nice), but there are key weaknesses (e.g., it describes incremental work), and it can significantly benefit from another round of revision. However, I won't object to accepting it if my co-reviewers champion it.

**Paper Topic And Main Contributions:**

This paper releases a medical Q&A dataset with 7K problems from the Chinese Pharmacist Examination. The authors then did a comprehensive analysis of the state-of-the-art large language models over this new benchmark. The results show most LLMs trained on general text do not answer well.


**Questions For The Authors:**

- GPT-3.5 v.s. ChatGPT: In most of the literature, the `text-davinci-003` model is usually called "GPT-3" while the `ChatGPT` model is usually called `GPT-3.5`.

**Reasons To Accept:**

- This paper releases a new Chinese medical Q&A dataset that builds a new benchmark for testing the performance of medical LLMs.

- Associated with the Q&A data, the medical explanations are also offered as a way to verify the correctness of knowledge stored in LLMs when making generations.

**Reasons To Reject:**

- The compared baselines are all not specifically in the medical domain. For instance, although ChatGLM-6B, BELLE, and ChatYuan were trained on Chinese corpus, they were not designed to answer medical questions. Surprisingly, GPT-4, without specific fine-tuning, yields over 75\% on the benchmark. It would be more convincing about the challenge of this benchmark if we could test more Chinese medical LLMs over it, e.g., a bunch of open Chinese medical LLMs is available at https://github.com/HqWu-HITCS/Awesome-Chinese-LLM#%E5%8C%BB%E7%96%97.


**Reproducibility:**

5: Could easily reproduce the results.

**Reviewer Confidence:**

5: Positive that my evaluation is correct. I read the paper very carefully and I am very familiar with related work.

---

> ### Author Rebuttal · Authors · 2023-08-23
>
> We would like to express our sincere gratitude for your meticulous review and valuable feedback on our paper. Your detailed observations offer us an opportunity to clarify certain aspects and provide a clearer perspective. We'd like to address each of your concerns.
>
> **Choice of LLM Baselines:**
>
> We appreciate your suggestion to assess domain-specific models like BenTsao and DoctorGLM using the resources provided. In our revised paper, we plan to include test results from additional Chinese medical Large Language Models (LLMs) to make our benchmarking more comprehensive. It's worth noting that many checkpoints were not publicly disclosed at the time our paper was originally submitted.
>
> We have already evaluated two popular medical LLMs—ChatGLM-Med and Huatuo-Llama-Med-Chinese—and found that they struggled with our multi-choice questions. The accuracy scores were as follows:
>
> - ChatGLM-Med in a zero-shot setting: 17.5
> - ChatGLM-Med in a one-shot setting: 10.1
> - Huatuo-Llama-Med-Chinese in a zero-shot setting: 18.5
> - Huatuo-Llama-Med-Chinese in a one-shot setting: 19.6
>
> These findings indicate room for improvement in these domain-specific models.
>
> **GPT-3.5 vs. ChatGPT Nomenclature:**
> We apologize for the oversight regarding the nomenclature of the GPT models. You are right; the naming convention we used can lead to confusion. We will correct this in the revised paper, adhering to the widely accepted nomenclature in the literature.
>
> **Incremental Work and Further Revision:**
> We understand your sentiments regarding the paper's merits and the areas it could benefit from further refinement. Our primary goal was to introduce a novel dataset and provide preliminary insights into its challenges. We take your feedback seriously and believe that another round of revision can significantly enhance the paper, addressing the mentioned weaknesses.
>
> In conclusion, we believe that with the suggested improvements and additional evaluations, our work will present a more solid contribution to the research community. We appreciate the time and expertise you invested in reviewing our paper and are committed to refining our work based on your valuable feedback.

---

### Official Review · Reviewer_xaZa · 2023-08-02

**Soundness:** 3

**Excitement:**

3: Ambivalent: It has merits (e.g., it reports state-of-the-art results, the idea is nice), but there are key weaknesses (e.g., it describes incremental work), and it can significantly benefit from another round of revision. However, I won't object to accepting it if my co-reviewers champion it.

**Paper Topic And Main Contributions:**

The paper introduces an important and novel dataset, ExplainCPE, in the specialized domain of medicine.

**Reasons To Accept:**

1. The creation of ExplainCPE, a medical dataset consisting of over 7K problems from Chinese Pharmacist Examination, is a significant contribution.
2. The comparison of different LLMs' performance, with GPT-4 outperforming others, provides valuable information about the strengths and weaknesses of current models in the medical domain.


**Reasons To Reject:**

1. A dataset paper that didn't propose a novel method.
2. EMNLP has an acceptance rate.

**Reproducibility:**

3: Could reproduce the results with some difficulty. The settings of parameters are underspecified or subjectively determined; the training/evaluation data are not widely available.

**Reviewer Confidence:**

2: Willing to defend my evaluation, but it is fairly likely that I missed some details, didn't understand some central points, or can't be sure about the novelty of the work.

---

> ### Author Rebuttal · Authors · 2023-08-23
>
> Thank you for your insightful feedback on our paper. We appreciate both the commendations and the critiques, and would like to address your concerns directly.
>
> **Nature of the Paper:**
> We understand the concern about the paper primarily being dataset-centric without proposing a novel method. However, we'd like to emphasize that the introduction of specialized datasets, especially in domains as critical as medicine, is pivotal for the advancement of natural language processing (NLP) techniques. ExplainCPE fills a gap in the NLP community by providing a resource that can help in understanding the intricacies of medical data.
>
> **Comparison of LLMs:**
> While the paper doesn't introduce a new model, the comparative study on how different LLMs, including GPT-4, perform on ExplainCPE offers invaluable insights. This analysis not only benchmarks the models but also directs future research towards the areas where they may be lacking, especially in specialized fields like medicine.
>
> **EMNLP Acceptance Rate:**
> While we acknowledge EMNLP's competitive acceptance rate, the unique nature and the value of ExplainCPE, combined with the thorough evaluation on prominent models, make a strong case for its consideration. Our work can act as a foundation for many downstream tasks and inspire novel methodologies tailored for medical NLP tasks.
>
> In conclusion, while the primary contribution of our paper is the dataset, its significance and potential implications for future research are immense. We hope the committee will recognize the potential impact of ExplainCPE on the community and the advancements it could drive in medical NLP tasks.

---

### Official Review · Reviewer_1fgk · 2023-08-04

**Soundness:** 3

**Excitement:**

2: Mediocre: This paper makes marginal contributions (vs non-contemporaneous work), so I would rather not see it in the conference.

**Paper Topic And Main Contributions:**

This paper proposes a Chinese pharmacist examination data set, which are mostly multiple choice problem. For each question, both the correct answer and the textual explanation are given. The authors use this data set to evaluate the performance of some GPT models, like ChatGLM, ChatGPT and GPT-4.

**Reasons To Accept:**

1. This paper proposes a new data set of Chinese Pharmacist examination. This data set includes not only the correct answer but also the explanations.

2. The authors evaluate on three popular GPT models, ChatGLM, ChatGPT and GPT-4.

**Reasons To Reject:**

1. The major contribution of this paper is the proposed dataset. However, the quality of this data set is not evaluated. For example, whether the label is correct, whether the explanation is accurate and coherent.

2. The human evaluation is conducted on the results of GPT models. An expert evaluation on the golden labels should be included.

3. The sources where the data set is collected should be described in detail.

4. It's more like a resource paper and may not be proper for the NLP application track.

**Reproducibility:**

3: Could reproduce the results with some difficulty. The settings of parameters are underspecified or subjectively determined; the training/evaluation data are not widely available.

**Reviewer Confidence:**

4: Quite sure. I tried to check the important points carefully. It's unlikely, though conceivable, that I missed something that should affect my ratings.

---

> ### Author Rebuttal · Authors · 2023-08-23
>
> Thank you for taking the time to review our paper and for providing feedback. We'd like to address your concerns point by point.
>
> **Quality of the Proposed Dataset & Human Evaluation on GPT Models vs. Expert Evaluation on Golden Labels:**
> The essence of the paper centers on introducing the new Chinese pharmacist examination dataset. We ensured during its formation that the data was gathered from trustworthy and authoritative sources. The dataset collection process is elaborated upon in Line 233-258. **Each official question's explanation is sourced from official examination solution books**, such as the one from [2017](https://www.amazon.com/%E8%8D%AF%E5%AD%A6%E7%BB%BC%E5%90%88%E7%9F%A5%E8%AF%86%E4%B8%8E%E6%8A%80%E8%83%BD-%E5%9B%BD%E5%AE%B6%E6%89%A7%E4%B8%9A%E8%8D%AF%E5%B8%88%E8%B5%84%E6%A0%BC%E8%80%83%E8%AF%95%E5%91%BD%E9%A2%98%E7%A0%94%E7%A9%B6%E5%A7%94%E5%91%98%E4%BC%9A/dp/B06ZZFK25X/), which is authored by the National Pharmacist Qualification Examination Questionnaire Research Committee. This ensures the accuracy and rigor of the explanations.**In addition to the official questions, we also collaborated with three doctoral students from Peking Union Medical College (all of whom have undergone standardized residency training). They manually reviewed 320 samples from the collected data to evaluate the completeness and accuracy of the label and explanations. The evaluation resulted in the 99.4%/99.0% accuracy rate, with 318/317 out of the 320 samples being deemed correct.** We will enhance the paper by incorporating this detailed evaluation of dataset quality. Such improvements can encompass both a statistical assessment of label correctness and a qualitative review of the coherence and accuracy of the provided explanations.
>
>
> **Sources for Data Collection:**
> The confidentiality and integrity of the data sources are crucial for the credibility of our dataset. We would like to extend the paper in the appendix to include a detailed section on the sources of our data collection, the methodologies used, and the precautions taken to ensure data fidelity.
>
> **Appropriateness for the NLP Application Track:**
> While the paper might seem resource-centric due to its focus on a new dataset, the evaluation of the dataset on popular GPT models makes it relevant to the NLP application track. We believe that introducing and evaluating new datasets is pivotal to pushing the boundaries of current models and driving research in specialized areas, like Chinese pharmacist examinations in this case. However, we respect the thematic guidelines of the track and are willing to make necessary amendments to better suit the scope.
>
> We believe that with the suggested improvements, our paper will stand as a valuable contribution to the community. We appreciate your insightful feedback and are eager to refine our work based on it.

---

### Meta-Review · Area_Chair_9YDD · 2023-09-19

**Recommendation:** 2

**Metareview:**

The paper proposes ExplainCPE, a new dataset focusing on the Chinese Pharmacist Examination. This dataset is unique in that it includes not only multiple-choice questions but also their corresponding textual explanations. Three prevalent GPT models (ChatGLM, ChatGPT, and GPT-4) are evaluated on this dataset. While reviewers appreciate the contribution of a new dataset and some preliminary evaluations, they also raise concerns regarding the quality of the dataset, the scope of the evaluations, and the paper's fit within the NLP application track. The quality of the dataset is not sufficiently validated. All reviewers agree that the creation of a new medical dataset specific to the Chinese Pharmacist Examination is a valuable contribution to the community. Questions have been raised regarding the correctness of the labels and explanations. The evaluation lacks a comparison with domain-specific LLMs, which would provide a better context for understanding the utility of the dataset. The paper should also provide more information on how the dataset was collected, which would help assess its reliability and usefulness. Considering both positive and important issues highlighted by the reviewers, this paper falls into the "Borderline Sound" category.

---

### Decision · Program_Chairs · 2023-10-07

**Decision:**

Accept-Findings

**Comment:**

The paper proposes ExplainCPE, a new dataset focusing on the Chinese Pharmacist Examination. This dataset is unique in that it includes not only multiple-choice questions but also their corresponding textual explanations. Three prevalent GPT models (ChatGLM, ChatGPT, and GPT-4) are evaluated on this dataset. While reviewers appreciate the contribution of a new dataset and some preliminary evaluations, they also raise concerns regarding the quality of the dataset, the scope of the evaluations, and the paper's fit within the NLP application track. The quality of the dataset is not sufficiently validated. All reviewers agree that the creation of a new medical dataset specific to the Chinese Pharmacist Examination is a valuable contribution to the community. Questions have been raised regarding the correctness of the labels and explanations. The evaluation lacks a comparison with domain-specific LLMs, which would provide a better context for understanding the utility of the dataset. The paper should also provide more information on how the dataset was collected, which would help assess its reliability and usefulness. Considering both positive and important issues highlighted by the reviewers, this paper falls into the "Borderline Sound" category.